# Caval Valve Implantation (CAVI): An Emerging Therapy for Treating Severe Tricuspid Regurgitation

**DOI:** 10.3390/jcm10194601

**Published:** 2021-10-07

**Authors:** Omar Abdul-Jawad Altisent, Rimantas Benetis, Egle Rumbinaite, Vaida Mizarien, Pau Codina, Francisco Gual-Capllonch, Giosafat Spitaleri, Eduard Fernandez-Nofrerias, Antoni Bayes-Genis, Rishi Puri

**Affiliations:** 1Department of Cardiology, Germans Trias University Hospital, 08916 Badalona, Spain; pau.codi@gmail.com (P.C.); fgualc@gmail.com (F.G.-C.); giosafat.spitaleri@yahoo.it (G.S.); nofrerias@gmail.com (E.F.-N.); abayesgenis@gmail.com (A.B.-G.); 2Department of Cardiothoracic and Vascular Surgery, Lithuanian University of Health Sciences, 44307 Kaunas, Lithuania; rimantas.benetis@kaunoklinikos.lt; 3Department of Cardiology, Lithuanian University of Health Sciences, 44307 Kaunas, Lithuania; rimantas.benetis@kaunoklinikos.lt (E.R.); rimantas.benetis@kaunoklinikos.lt (V.M.); 4Department of Cardiology, Cleveland Clinic, Cleveland, OH 44195, USA; purir@ccf.org

**Keywords:** tricuspid regurgitation, Transcatheter Valves Therapies, bicaval valve implantation, review

## Abstract

Severe tricuspid regurgitation remains a challenging heart-valve disease to effectively treat with high morbidity and mortality at mid-term. Currently guideline-directed medical treatment is limited to escalating dose of diuretics, and the rationale and timing of open-heart surgery remains controversial. Emerging percutaneous therapies for severe tricuspid regurgitation continue to show promising results in early feasibility studies. However, randomized trial data is lacking. Additionally, many patients are deemed unsuitable for these emerging therapies due to anatomical or imaging constraints. Given the technical simplicity of the bicaval valve implantation (CAVI) technique compared to other transcatheter devices, CAVI is postulated as a suitable alternative for a wide variety of patients affected with severe+ tricuspid regurgitation. In this review we illustrate the current evidence and ongoing uncertainties of CAVI, focusing on the novel CAVI-specific devices.

## 1. Introduction

Severe symptomatic tricuspid regurgitation (TR) is associated with poor short- to medium-term clinical outcomes, representing a leading cause of moderate-to-severe heart valve disease in developed countries [1,2]. Medical treatment and tricuspid valve (TV) surgery are the currently accepted therapies to treat severe symptomatic TR. However, despite TV surgical volume steadily increasing in the US, in-hospital mortality has remained static at 9%, suggestive of the fact that TV surgery is typically offered too late in the course of disease [3,4]. Several percutaneous devices have emerged during last decade to treat severe TR, with promising results in early feasibility studies, with pivotal randomized trials ongoing [5]. However, owing to its complex anatomy and challenges with peri-procedural imaging, many patients are still deemed unsuitable for these emerging percutaneous TV therapies that include edge-to-edge repair, direct annuloplasty and orthotopic valve replacement.

Caval valve implantation (CAVI) emerged initially as an alternative therapy for patients deemed as having ‘no other options’ for treating their severe symptomatic TR; such patients often afflicted with concomitant hepatic congestion and right heart failure [5]. Given the emerging challenges and applicability of many emerging percutaneous TV therapies to the broader cohort of severe TR candidates, the simplicity of CAVI underscores its attractiveness as an effective treatment option for many severe TR patients. CAVI has subsequently rapidly evolved with the development of dedicated devices adapted to the specific anatomy of the caval venous system [6,7]. The objective of this review is to illustrate the current evidence and ongoing uncertainties of CAVI, advantages/disadvantages compared with other percutaneous TV devices, CAVI-related planning and technical aspects, and post-implant care of CAVI recipients.

## 2. Context

Severe TR is a complex valve disease based on heterogeneous TV anatomy along with variations in its etiology and clinical presentation. Secondary TR, usually associated with pulmonary arterial hypertension or left heart disease, is manifested by significant right ventricle (RV) dilatation in the longitudinal direction (spherical deformation of the RV) and/or tricuspid annular dilatation towards the RV free wall. These structural alterations result in tethering of the TV leaflets resulting in coaptation gaps. Conversely, in isolated TR (without significant pulmonary hypertension) the RV typically has a normal length and TR is mainly caused by tricuspid annular dilation of the right atrium (RA), mainly in the presence of atrial fibrillation [8].

With the abovementioned anatomical and pathological changes in mind, a range of transcatheter strategies have been developed for targeting severe TR according to the underlying mechanism of disease. Briefly, annuloplasty devices are designed to reduce annular dilatation and help indirectly restore leaflet coaptation; edge-to-edge repair aims to create a leaflet tissue bridge across the regurgitant orifice; direct orthotopic valve replacement can be used in patients with large annuli and significant coaptation gaps where edge-to-edge repair and/or annuloplasty would unlikely to be effective (Table 1) [9,10,11,12,13,14,15]. However, despite this varied range of new devices, the majority of the overall severe TR population are ultimately deemed anatomically unsuitable for currently tested transcatheter therapies [14,16]. The reasons for this are partly explained by anatomical complexity/variability, which makes it difficult for a single device to adapt to the specific characteristics of each patients anatomy. Additionally, reproducible and interpretable periprocedural imaging remains challenging. Consequently, procedural times are typically long, and technologies are limited to highly trained/skilled operators in high-volume centers, thus restricting generalizability to the broader interventional cardiology community.

By contrast, compared with other tricuspid devices, CAVI procedures have some inherent advantages owing to the relative simplicity of the caval anatomy relative to the TV and right heart (Table 1) [17]. Anatomical (and peri-procedural imaging) requirements for CAVI are less strict. CAVI with new dedicated-devices may thus be a reasonable option for the majority of patients screened out for edge-to-edge repair/orthotopic valve implantation/annuloplasty, or be considered a rescue technique in cases with failure of other devices. CAVI has a TAVI-like workflow, involving computed tomography (CT) planning with reproducible procedural planning and steps [6]. Echocardiographic imaging during the procedure or general anesthesia can be helpful early in the learning curve but by no means mandatory. Consequently, the procedure is relatively simple and faster than other transcatheter TV devices, enabling both the learning curve and the generalization of the procedure to novice operators/lesser experienced centers. CAVI may be a good option for patients with pacemaker lead-induced TR because the device does not interfere with the tricuspid valve. For all these reasons, the potential CAVI population will likely represent a reasonably high proportion of the overall severe TR population [6]. Hence, as data continue to emerge, CAVI should not necessarily be seen as the last alternative in patients without other options; rather as a potential promising alternative for a broad range of severe TR patients at high or prohibitive surgical risk.

### 2.1. CAVI: Concept

CAVI is intended to alleviate congestive signs for patient affected by severe to torrential TR considered inoperable/high-risk for conventional cardiac surgery. The concept centers in the heterotopic placement of a valve in the inferior vena cava (IVC) and another in the superior vena cava (SVC), at the cavo-atrial junctions [6,18]. The rationale is that preventing regurgitant flow within the IVC in turn reduces liver congestion, improves liver and renal function with subsequent reductions in abdominal congestion, ascites, and peripheral edema. In addition, ameliorating caval regurgitant volume ultimately increases RV stroke volume into the pulmonary circulation improving cardiac output. At the chronic phase, reduction in peripheral congestion decreases the risk of cardiac cirrhosis, also reducing right heart overload promoting a degree of reverse RV remodeling, and possibly even reductions in tricuspid annular dilatation, and thus the severity of TR in some cases. A global improvement of patient functional status, symptoms of congestion/dyspnea and ultimately reductions in hospitalization rates are expected. Whether life expectancy can be improved remains to be proven [18,19].

### 2.2. CAVI: Anatomical Considerations

The SVC is 7.1 cm ± 1.4 long and is formed by the union of the right and left brachiocephalic trunks (also called innominate); it returns the blood from the upper half of the body to the heart. It courses along the right middle mediastimun, with the trachea and ascending aorta on its left and drains over the anterior roof of the RA. SVC is often irregular in shape. The Azygos vein is a major tributary and drains posteriorly into distal SVC. There are no other major tributaries, however SVC has several patterns of venous collaterals. Neither the SVC nor the brachiocephalic trunk contains valves. They do give rise to the internal, external and subclavian jugular veins, and their valves are usually located at the entrance to the subclavian and jugular veins. These vein valves can hinder the progression of catheters or wires. Other notable anatomical structures of the RA, such as the sinus node, are located in a lower level, in an area where CAVI devices should not be implanted (Figure 1 and Figure 2) [20,21]. Persistent left SVC is the most common congenital thoracic venous anomaly. Other congenital or acquired anomalies such as partial anomalous pulmonary venous return, SVC aneurysm or stricture may cause difficulty in CAVI procedures. 

The IVC is typically larger than the SVC, it returns the blood from the inferior part of the body to the heart. Its junction with the RA is at its inferior and posterior part, close to the inter-atrial septum. Therefore, junction of the RA-IVC and RA-SVC are not coaxial. In the anterior part of the junction between IVC-RA there is a valve (Eustachian Valve) that it is variable in size and it extends to the lateral/right part of the vein. The Eustachian valve can be easily viewed by transesophageal echocardiography (TEE) in the bi-caval view. Angulations around the RA-IVC and above-below the hepatic veins and RA-junction may be present, which could pose hindrance to CAVI (Figure 1). The diaphragmatic orifice of the IVC is quadrilateral, related to the phrenic center and crossed by the IVC, which it is attached to its edges, and by some right branches of the phrenic nerve (sensitive ones). Importantly, an exaggerated overexpansion of a device at this level could compress branches of the phrenic nerve and cause phrenic pain (shoulder pain) or hemi-diaphragm paresia. Superior hepatic veins are usually composed by a large right, left and middle veins. They have no valve. Distance between the IVC-RA junction to supra-hepatic veins confluence is variable, but it is usually between 5 to 15 mm (Figure 1 and Figure 3) [22]. Congenital IVC variants are present in 4% of patients. The most common anomalies are the absence of IVC, duplication of the IVC (usually in infrarenal segments), left-side IVC, anomalous continuation of the suprarenal IVC as the Azygos or Hemiazygos vein, retrocaval ureter, IVC web formation (complete or fenestrated membrane in the intrahepatic IVC segment) and extra-hepatic porto-caval shunt (Abernethy malformation).

### 2.3. CAVI: Evidence and Devices

The chief concern regarding CAVI is that there is still scarce data published regarding its effectiveness. However, similar to other transcatheter tricuspid devices, initial first-in-man studies also showed promising results in terms of feasibility, safety and device performance (Table 2). To date, 4 different CAVI approaches have been investigated: a non-dedicated device (Edwards Sapien/Direct-flow plus pre Z stenting); and three devices specifically designed for caval anatomy, the TricValve^®^ system, the Tricento^®^ system, and the Trillium^TM^ system.

### 2.4. Non-Dedicated CAVI Devices: Early CAVI Studies

Lauten and Figulla initially studied the impact of CAVI to improve TR in an animal model using a self-expandable valve stent. The authors found that the procedure were feasible in 100% of cases, showing a significant decrease in tricuspid regurgitation volume into the cava, coupled with an increment in the cardiac output in both the acute and chronic phase [30]. The same authors reported the first-in-man series of CAVI for treating patients with severe symptomatic TR [25]. This early study included 25 patients deemed at prohibitive risk for open-heart surgery. The majority of patients were treated with only one valve (within the IVC) via femoral access using a non-dedicated device: the balloon expandable Edwards Sapien 29 mm with prior Z-stenting of the IVC. The authors demonstrated a high degree of procedural success with 92% of successful implants, a reduction of regurgitation volume into IVC in all patients (on basis of reduction of v-wave in the IVC), and 84% of patients showed an improvement of at least 1 New York Heart Association heart failure class. In-hospital mortality was 24%, but the discharged patients demonstrated continued valve function during follow-up. This relatively high in-hospital mortality, however, highlighted the inherent risk of the patients included in the study. Main studies evaluating non-dedicated CAVI devices are summarized in Table 2 [7,24,25,26,27,28,29,30].

One of the major concerns of using a non-dedicated device for CAVI is prosthesis migration or embolization, some of which are detected at a late stage. This was the main issue of the TRICAVAL study, which randomized 28 patients to CAVI with a non-dedicated device vs. medical therapy (14 patients each) [26]. In this study there were 2 delayed migrations and 2 embolizations that forced the closure of the study for safety reasons [26]. This reflects the challenges of using non-dedicated devices in the venous circulation. Other main limitation of using non-dedicated CAVI devices is prosthesis size. Vena cava tends to be quite large in severe TR patients, and usually non-dedicated devices are not large enough for the caval dimensions.

Another unresolved question is whether one prosthesis (only for the IVC) is better than two (treating the SVC as well). Initial studies used only one prosthesis in IVC and showed improvements in the signs/symptoms of congestion. The implantation of a single IVC valve likely reduces the risk of device migration than the implantation of 2 valves. However, it seems that two valves present superior hemodynamic performance than 1 [31].

## 3. Dedicated Devices

### 3.1. TricValve^®^: Evidence, Current Studies, and Future Trials

Special access program for compassionate use of the dedicated TricValve^®^ system has been granted in 11 countries. To date, 47 patients have been treated within this program, with an implantation success of 98%, with only 1 out of 47 patients suffering from device embolization requiring surgical correction. So far, in hospital mortality is 0% and 30-day mortality is 4% (Table 2), comparable with other tricuspid percutaneous devices.

Besides the global compassionate use program, there are currently 2 ongoing trials with specific protocols assessing the feasibility, safety and performance of the TricValve^®^ system: TRICUS STUDY (NCT03723239), an early feasibility first-in-man trial; and TRICUS-EURO (NCT04141137), a CE mark trial. End-points also include assessment of the functional status, exercise capacity and quality of life. The 2 trials have single arm-an open label design and have completed patient enrollment. TRICUS included 9 patients from Lithuania, and TRICUS-EURO included 35 patients from Spain and Austria. Table 3 shows the clinical eligibility criteria for these 2 trials. Definitive results of these studies are expected to be published by the end of 2021.

### 3.2. TricValve^®^

Currently, TricValve^®^ is the only CAVI device with CE mark approval obtained in May 2021. The TricValve^®^ implantation system consists of 2 TricValve^®^ self-expanding nitinol structures with leaflets made out of bovine pericardium, one designed for the SVC and the other for the IVC–pre-mounted on a 27.5 F TricValve^®^ Delivery System. The bioprosthetic leaflets are processed with anti-calcification treatment as well as chemical dehydration for preloading in the delivery system (Figure 4A). The bioprosthesis is specifically designed to adapt to the anatomic features of the cava, to minimize the risk or migration and para-valvular leaks, and it is available in 3 different sizes for the SVC (25, 29 and 33 mm) and 4 for the IVC (31, 35, 41 and 45 mm). Its caval fixation is based on stent design, radial force, and the degree of oversizing during implantation. The SVC device has a long skirt to prevent para-valvular leak covering the ½ of the inferior portion of the belly (Figure 4A). It has a high radial force within the belly portion of the device to fixate and seal, as well as within the valve segment to eliminate the risk of valve deformation and malfunction. The crown of the superior prosthesis has low radial force for valve stabilization and for improving alignment with the SVC. The inferior prosthesis has a short skirt to prevent hepatic vein occlusion, covering only the first 20 mm of the proximal edge. It has also a high strength nitinol frame in its proximal segment, where the valve is placed (Figure 4A), The inferior or anchoring segment of the inferior prosthesis has low radial force with large diameter to increase contact area with soft pressure upon the caval wall [6].

### 3.3. TricValve^®^ Planning

Pre-implantation planning is based on clinical, hemodynamic and anatomical features. The system is designed for patients with at least severe TR. In summary, the TricValve^®^ studies only included patients severely symptomatic (NYHA functional class III or IV or with previous admission for heart failure during last 12 months) despite optimal medical therapy, and deemed at high or prohibitive surgical risk for conventional tricuspid surgery (Table 4). Pacemaker leads were not a contraindication. However, those with severe pulmonary hypertension (systolic pulmonary pressure >65 mmHg) or severe impaired right ventricular function (i.e., TAPSE < 13 mm) were contraindications. TricValve^®^ requires a significant tricuspid V-wave to ensure adequate valve leaflet motion. Usually, a tricuspid V-wave is large in patients with severe TR. However, the tricuspid V-wave can be low in some circumstances even with severe TR, particularly if the RA is very large, or if right ventricular function is depressed, particularly when the intravascular volume is depleted. Tricvalve^®^ should not be recommended if v-waves are <15 mmHg measured during a euvolumic setting.

Anatomically, bioprosthesis sizing requires accurate caval measurements on CT scanning to minimize the possibility of any safety issues and suboptimal performance results. Contrast-enhanced CT of the chest and abdomen, performed 70 to 85 s after injecting contrast agent into a peripheral vein is recommended for an uniform enhancement of both, SVC and IVC. An oversizing degree of 10 to 40% within the belly part for the SVC prosthesis and in the proximal part of the IVC prosthesis is desirable for optimizing fixation and seal. Table 4 shows the main anatomical references and the prosthesis valve sizing recommendations for the SVC and IVC. For the SVC, critical references are the diameter at the level of the confluence with the innominate trunk, the diameter at the level of pulmonary artery, diameter at the cavo-atrial junction, and the distance between these points. A minimum length of 50 mm between cavo-atrial junction and innominate confluence is needed for the SVC. The target position is to place the belly part of the SVC prosthesis, which has a long skirt covering the inferior half of the belly, at the level where the right pulmonary artery crosses the SVC. There should be caution in patients with short and tapered SVC, because risk of migration is higher in such anatomies. In these instances, a higher implant technique is recommended by placing the belly part of the SVC prosthesis between the confluence and the right pulmonary artery level (Figure 2). However, the belly part should not be positioned at the level of the confluence because of the risk of innominate occlusion. A high implantation technique is also recommended in patients with pacemaker or implantable cardioverter-defibrillator leads for minimizing the risk of migration (Figure 2).

For the IVC prosthesis, the critical points are diameters at the level of the junction with RA, below and above the hepatic veins, and the distance between supra-hepatic veins and cavo-atrial junction. Target position is to implant the proximal edge between the junction and the confluence of the supra-hepatic veins, entering into the atrium between 5 to 12 mm depending on the anatomy. The risk of peri-valvular leak increases if the proximal edge of the device enters within into the atrium more than 15 mm, since only the first 20 proximal mm of the device are covered by the skirt (Figure 3 and Figure 4). Special caution is required in patients with a short distance between the supra-hepatic veins and the cavo-atrial junction, in patients with significant differences in diameter above and below the hepatic veins, or in patients with significant angulations between the cavo-atrial junction. In these cases, the contact between the device and the caval wall is less and this could affect device stability. To increase the attachment area, the valve should be positioned as low as possible (entering only 0–5 mm into the atrium). A very slow deployment is also recommended.

### 3.4. TricValve^®^ Procedural Steps 

The implantation requires 3 access sites (option of 4th): 1 right common femoral vein (27.5F) for device deployment and 2 left common femoral vein (5F and 7F) for pigtail control injections and a Swan-Ganz catheter. Optionally a 4th left basilic vein access (5F) can be useful for better identifying the confluence between innominate and SVC. Pre-closure with 1–2 Proglides in the right common femoral vein can be useful for hemostasis, but not mandatory. A Swan-Ganz catheter should be placed at the level of right pulmonary artery for reference. Using the pigtail (+ a multipurpose catheter within the innominate vein), an angiogram of the SVC with special attention to the confluence and the distance between the innominate vein ostium and the SVC at the level where the right branch of the pulmonary artery crosses it is recommended for planning the deployment (Figure 5A,B). A stiff wire placed in the subclavian or internal jugular vein is recommended for valve deployment. There are usually venous valves at the entrance to the subclavian or jugular vein. Careful handling with a multipurpose catheter may be necessary to cross these valves, often via simple catheter rotation as opposed to pushing. Deployment is undertaken by rotating the device’s roulette clockwise. Deployment should be performed slowly, starting in a high position and pulling down the system until it arrives at the target position (Figure 5C). It is not recommended to push the device for avoiding a venous wall pinching. The system is retrievable until the 80% is deployed. If we need to implant the device in a higher position we should re-sheath the system completely and re-start again in a higher position. Being a self-expanding device with a lot of radial force, the device trends to come out of the delivery system, so in general it is necessary to maintain tension during the deployment. Once the belly part of the prosthesis is deployed the systems remains very stable and we can release the tension. It is important to be sure that all 3 hooks attached to the device are released. Before removing the delivery system out of the patient, one should re-sheath it completely to avoid vascular complications. 

For the IVC deployment, a venogram of the hepatic veins centered in the confluence with inferior cava and its junction with the RA is obtained, and serves as a reference for deployment (Figure 5D). The IVC prosthesis is deployed in a similar way to the SVC prosthesis, starting in a high position, into the right atrium, and pulling down the system at the same time that we are deploying the valve until our target position (proximal edge of the prosthesis landing in between the right atrium and supra-hepatic vein confluence) (Figure 5E). Depending on the anatomy or prosthesis size, a low implantation is typically recommended (as explained above). A TEE or transthoracic echocardiography can be useful, as the entrance or junction between IVC and right atrium is usually very well defined in the bi-caval view or sub-costal view. If echocardiographic images are available, a measure of the mm of the valve entering into the RA can be obtained during the deployment, as well as the presence of any leak (Figure 5F). The system is retrievable until 80% of the prosthesis is deployed. At this point the system remains very stable, then a release of tension is recommended for avoiding a jump effect when the knocks are untied. It is important to confirm that all 3 hooks attached to the device are released. The delivery system must be fully re-sheathed before removing it in order to avoid vascular complications.

### 3.5. Post-Implantation Care

Compared to percutaneous left heart procedures, CAVI is typically quite well hemodynamically tolerated during the procedure. However, following implantation there is an abrupt change in physiology characterized by a significant increase in right ventricular afterload due to amelioration of retrograde flow to the cava caused by the newly competent bi-caval valves, an acute increase in RA pressure, and increase in cardiac output if right ventricular function is preserved. This is coupled with significant systemic decongestion, with favorable effects especially seen hepatically. The individual response to these changes is variable and will largely depend up on baseline right ventricular function and its ability to adapt to increased afterload, baseline pulmonary arterial pressures, and left ventricular function.

Ventricularization of RA pressure is inevitable to a certain extent, with the concomitant risk of provoking further atrial arrhythmias despite the fact that the majority of these patients already have atrial fibrillation (Figure 6). In patients with preserved right ventricular function the acute increase in right ventricular stroke volume is usually well tolerated, however, in some patients may cause an acute increase in pulmonary arterial pressures and eventually, pulmonary congestion specifically if the left heart is not well compensated (Figure 6). In these cases the administration of pulmonary vasodilatators and diuretics could be useful initially.

Special attention is required inpatients with baseline right ventricular dysfunction, because the acute increases in right ventricular afterload may cause right ventricular failure. Usually this appears within the first 24 h, manifesting as systemic hypotension and oliguria. The response to low dose inotropes is generally good, with a weaning process instituted over 48 hrs. Prosthesis malfunction can be observed in patients with baseline low tricuspid v-wave as a minimum backflow pressure is required for proper prosthesis leaflet motion. 

Shoulder pain, likely caused by compression of sensitive branches of phrenic nerve at the level of diaphragm, is a common and usually benign and self-limited symptom following the procedure. The pain seems neuropathic in origin, and it is distributed in the dermatome of the phrenic nerve [32]. In our experience, only a small proportion of patients (around 5%) describe persistent pain over 3 or 4 weeks resistant to conventional/opioid analgesia. In these cases, gabapentin plus anxiolytics can be useful. 

A temporary platelet count decrease is also frequently observed during the first few days after post-procedure, and may be related with some chemical component of the prosthetic leaflets [33]. This adverse event, however, is usually benign, and typically the patient’s suppressed baseline platelet count recovers within the first week post-procedure.

There is an inherent risk of device migration since they are fixed upon a non-calcified venous vasculature. Migration usually appears within the first 48 hrs. An adequate anatomical assessment (already described above), resulting in a 10% to 40% degree of oversizing, coupled with an accurate and slow deployment technique reduces the risk of migration. Embolization is rare and in general is a consequence of errors in device sizing or secondary to non-accurate implantation technique. To date, there is no case documented of late post-procedural embolization. With the current valve sizes available, the device can be implanted in the vast majority of patients with minimum risk of device migration/embolization. 

### 3.6. Longer-Term Management

The improvement of liver congestion may require a change in the dosage required of vitamin K antagonist (VKA) in patients receiving this therapy, increasing the risk of bleeding due to VKA overdose. There is an inherent risk of valve thrombosis or dysfunction because the low pressure of central venous system, however these issues still require further analysis. Despite TricValve^®^ studies only recommending oral anticoagulation for 3 months, most of patients are already systemically anticoagulated due to the frequent presence of concomitant atrial fibrillation. The reduction in hepatic congestion also translates into the reduction in diuretic needs, and consequently possible improvement of renal function.

### 3.7. TRINCENTO^®^

#### Concept, Imaging, Anatomical Consideration and Evidence

The Tricento^®^ prosthetic valve is a self expandable bio-prosthesis consisting of a stent graft spanning from the inferior to the superior vena cava and a lateral bicuspid valve element. It is positioned within the right atrium and prevents bicaval backflow. The system is made from porcine peri cardium and Nitinol support structures. The covered stent has a maximum length of 13.5 cm. Anchoring is achieved by oversizing the stent elements in the area of overlap of stent and caval veins. This overlapping area also functions as a sealing zone that prevents blood from flowing back into the cava during systole (Figure 4B). Its delivery system uses a 24F stent graft catheter, designed to advance atraumatically into right atrium via femoral access, to be implanted in the SVC/right atrium and IVC in a controlled manner [7]. 

Tricento^®^ is intended for patients with severe TR deemed inoperable, who present with significant tricuspid V waves and preserved right ventricular function. Pre-implantation planning also includes an accurate CT imaging of the central venous anatomy and size. The prosthesis is currently customized on the basis of patient anatomy of the landing zone, Azygos and hepatic vein confluence [34,35]. 

There remains scarce data regarding the safety, performance, efficacy or durability of the Tricento^®^ Valve system (Table 2). The totality of evidence is currently summarized by case reports and, to best of our knowledge, there is not an ongoing prospective study assessing this device [36,37]. However, particularly in patients with challenging anatomies, there may be some benefit from a customized device such as Tricento^®^. 

### 3.8. TRILLIUM™

The Trillium™ (Innoventric Ltd., Ness-Ziona, Israel) is a cross-caval stent graft with multiple covered fenestrations arranged circumferentially around its RA portion. The covered fenestrations allow inflow of blood from the venous system into the RA and prevent backflow into the venous system. The Trillium™ is delivered using a 24 French delivery system in a strictly fluoroscopic procedure with transfemoral venous access. 

The main characteristics of the device include the multiple circumferential valves, which simplify the rotational positioning of the device, increase the total inflow area, and provide better safety of the technology. Positioning the device over existing pacemaker or implantable cardioverter-defibrillator leads is feasible. Additionally, the Trillium™ has a designated skirt for blocking IVC backflow without blocking supra hepatic vein inflow in patients with very small distance between the supra hepatic veins and the RA cavity (Figure 4C). Currently, the Trillium™ system is in the midst of its first in human clinical trial (NCT04289870), with over 10 patients treated so far in Germany, Belgium and Israel.

## 4. Conclusions

Initial data suggest that CAVI procedures with specific-designed devices are feasible and safe for treating severe TR. Procedures with CAVI-specific device are relatively simple and predictable, requiring less anatomical pre-requisites and interventional skills than other percutaneous tricuspid devices. Thus, these devices are a promising alternative for a wide range of patients affected of severe tricuspid regurgitation. However, theirs favorable effects from a safety and efficacy perspective still needs to be corroborated by the currently ongoing pivotal trials, and ultimately in a pivotal randomized trial against optimal medical therapy.

## Figures and Tables

**Figure 1 jcm-10-04601-f001:**
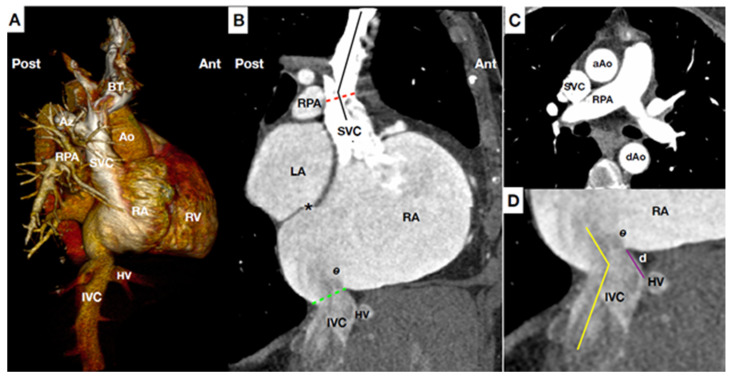
Main anatomical landmarks for CAVI. (**A**) Computed tomography (CT) scan reconstruction of central venous system, right heart and pulmonary artery at lateral plane. (**B**) CT para-sagittal plane at the level of the vena cava. Superior vena cava (SVC) is usually 7.1 cm long, and has slight angulation (black line), usually at the level of right pulmonary artery (RPA) (Red line). Of note, the junction of SVC-right atrium (RA) and junction IVC-RA are not in the same anterior-posterior plane. Special caution applies to patients with short distance between the IVC-RA junction (green line) and inter-atrial-septum (*) owing to the risk of septal perforation during inferior caval prosthesis delivery. Eustachian valve (e) varies in size, may be very prominent. (**C**) CT transversal plain at the level of RPA. Notable structures (i.e., aorta, trachea, pulmonary artery, etc) are adjacent but not attached to SVC minimizing risk of complications during CAVI. (**D**) CT at para-sagittal plane of the IVC-RA junction showing the distance between hepatic vein (HV) confluence and IVC-RA junction (purple line). Higher distance at this level helps to stabilize the devices owing that it increases the contact surface between device-vein wall. The IVC-RA junction (yellow line) usually presents with a greater angulation than SVC-RA junction. Ao: Aorta. AZ: Azygos vein. BT: Brachiocephalic trunks (Innominate). dAo: descending Aorta. d.: distance between hepatic vein confluence and inferior vena cava-right atrium junction. HV: Hepatic veins. IVC: Inferior vena cava. LA: Left atrium. RA: Right atrium. RPA: Right pulmonary artery. RV: Right ventricle. SVC: Superior vena cava.

**Figure 2 jcm-10-04601-f002:**
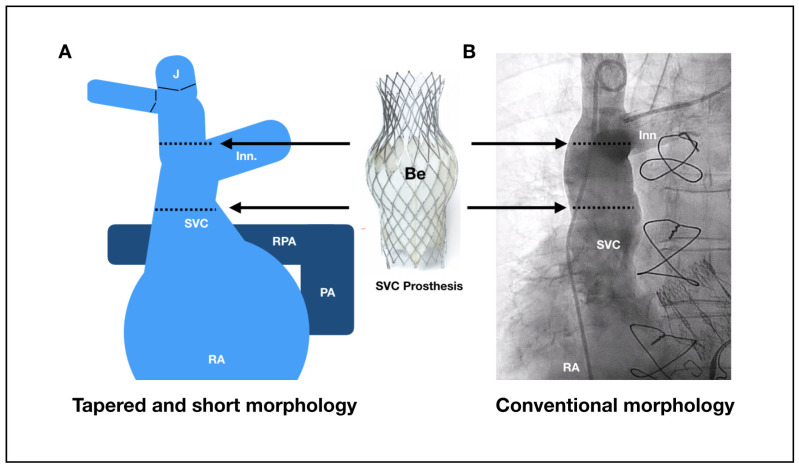
Anatomical diagram and angiographic view of superior vena cava. (**A**) Anatomical diagram of a tapered and short superior vena cava (SVC). The morphology of the superior vena cava showed in the diagram increases the risk of prosthesis migration/embolization into the right atrium (RA). A high implantation is thus recommended to prevent it (target position is to implant the belly part of the SVC prosthesis at the level of the innominate plane without obstructing the ostium). (**B**) Angiographic view at posterior-anterior plane showing a SVC with conventional morphology. In these cases target position is to place the belly part of the SVC prosthesis at the level where the right pulmonary artery crosses the SVC. Be: Belly part of the superior prosthesis. J: common Jugular vein. Inn: Innominate vein. PA: Pulmonary Artery. RA: Right Atrium. RPA: Right Pulmonary Artery. SVC: Superior Vena Cava. T: Tricuspid Valve.

**Figure 3 jcm-10-04601-f003:**
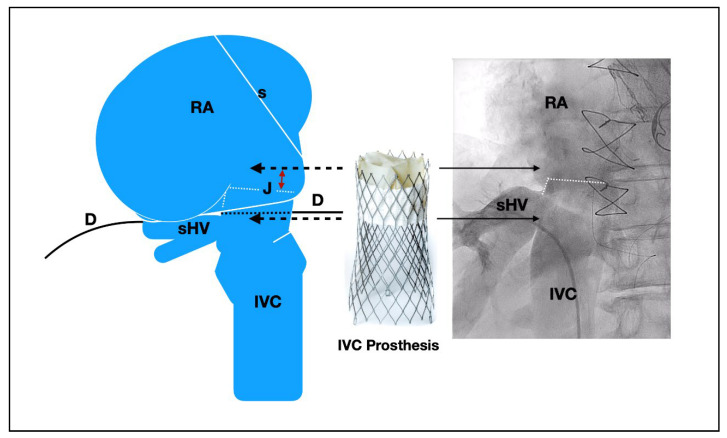
Anatomical diagram and angiographic view of the inferior vena cava. Anatomical diagram and angiographic view at posterior-anterior plane of the inferior vena cava (IVC), cavo-atrial junction (J) and right supra-hepatic vein (sHV). Ideally the target position for TricValve^®^ implantation is to place the proximal edge of the inferior prosthesis between 5 to 12 mm into RA (red double arrows). The proximal part of the inferior prosthesis has high radial force and is partially covered by a skirt (only covers the 20 mm proximal of the prosthesis). Distance between cavo-atrial junction and sHV is usually around 5 to 15 mm. Diaphragm (D) and branches of the phrenic nerves are the notable structures that could be damaged during the implantation of the inferior vena cava prosthesis. Important angulations between IVC, sHV and cavo-atrial junction hinder the prosthesis deployment. In such cases, a very slow deployment and low implantation is recommended. Be: Belly part of the superior prosthesis. D: Diaphragm. Inn: Innominate vein. IVC: Inferior Vena Cava. J: cavo-atrial Junction. PA: Pulmonary Artery. RA: Right Atrium. RPA: Right Pulmonary Artery. S: Intra-atrial septum. SVC: Superior Vena Cava vein.

**Figure 4 jcm-10-04601-f004:**
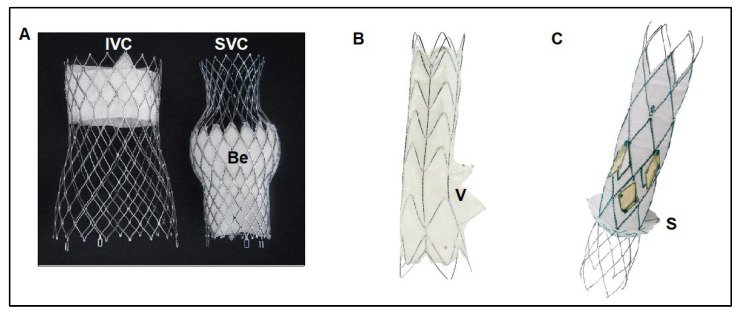
Current dedicated CAVI devices. (**A**) The TricValve^®^ device consists of 2 self-expanding nitinol valvular stent frame devices specifically designed for the cava, one for the SVC and another for the IVC. The SVC device has a belly portion (Be) with high radial force to fix and seal the device into the SVC, has a long skirt that covers the 1/2 inferior parts of the device for preventing para-valvular leak. The valve is placed in the inferior part of the device. The IVC device has a short skirt only covering the first 20 mm of the proximal edge for preventing hepatic vein occlusion, and high radial force at the valve level for fixing the device and preventing valve distortion or malfunction. (**B**) Tricento^®^ device is a customized device adapted to the anatomy of each particular patient. Device consists of a self-expanding covered stent frame placed into the SVC, right atrium and IVC with an opening valve (V) facing the tricuspid plane. (**C**) Trillium™ device is a cross-caval stent graft with multiple covered fenestrations arranged circumferentially around its right atrium portion. Trillium™ has a skirt at its inferior part to seal without occluding hepatic veins. Be: Belly part of TricValve^®^ superior prosthesis. IVC: Inferior Vena Cava. SVC: Superior vena cava. S: Skirt of Trillium™ system. V: Valve of Tricento^®^ system.

**Figure 5 jcm-10-04601-f005:**
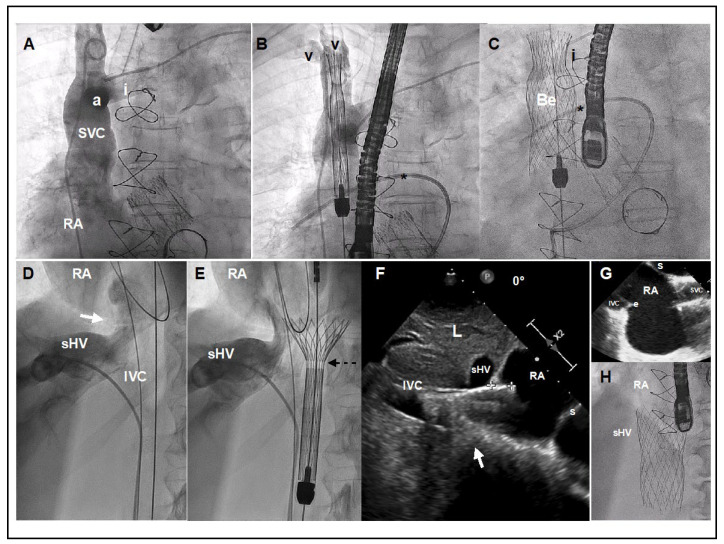
TricValve^®^ implant details. (**A**) Venogram of the SVC with careful visualization of the innominate vein (i). The venogram was performed using a dual injection technique, 1 using a pig-tail through left femoral access, and another using a multipurpose through left basilica vein access. (**B**,**C**) Target position is to implant the belly part of the device (segment with higher radial force) at the level of pulmonary artery plane or between pulmonary artery plane and confluence with the innominate vein. A Swan-Ganz catheter placed into the right pulmonary artery and multipurpose catheter (or leads if patient has a pacemaker) may be a helpful reference. (**D**) Venogram with careful visualization of the right supra-hepatic vein confluence, IVC and junction with the right atrium (white arrow). (**E**) The first 20 mm of the proximal edge of the inferior valve is covered by a skirt to prevent occlusion of the hepatic vein. A radiolucent mark shows where the skirt of the valve ends (broken arrow). (**F**) TEE at 0º, showing the inferior prosthesis during deployment entering 11 mm into RA (between crosses). White arrow shows the cavo-atrial junction. Optimally positioning requires only 5 to 12 mm of the proximal portion should enter into the RA. (**G**) TEE bi-caval view after deployment of the superior and inferior prosthesis. (**H**) Angiographic view of the inferior vena cava prosthesis immediately after deployment. a: Azygous confluence. Be: Belly part of the superior vena cava prosthesis. e: Eustachian valve. I: Innominate vein. IVC: Inferior Vena Cava. L: Liver. RA: Right Atrium. sHV: superior Hepatic Vein. s: Septum. V: Vein valves of Subclavian and Jugular veins. Star: Right Pulmonary Artery with Swan Ganz.

**Figure 6 jcm-10-04601-f006:**
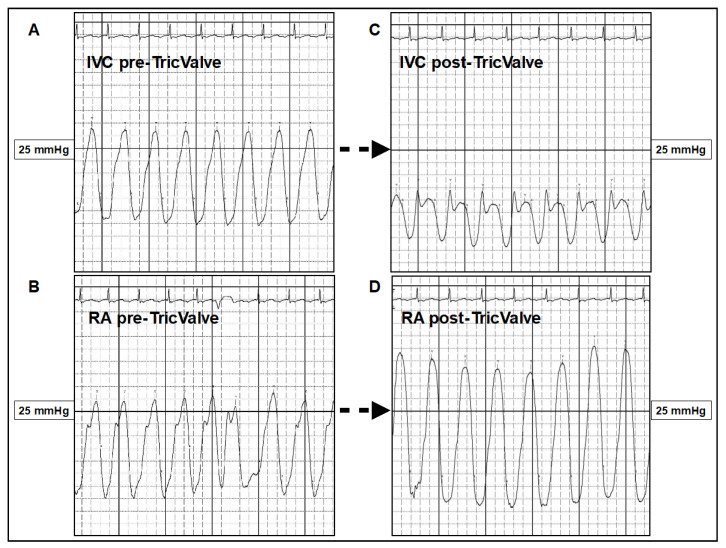
Ventricularization of the RA following TricValve^®^ implantation. (**A**,**B**) Baseline, IVC pressure curve is 18 mmHg (and v-wave is 28 mmHg) and in the RA is 17 mmHg (V-wave is 27 mmHg). (**C**,**D**) Following TricValve^®^ implantation there is a clear decrease of the caval pressures with a corresponding change in the curve morphology (**C**). As a consequence of the acute afterload increase, there is an increase in the RA pressure with ventricularization of the curve morphology in the acute phase (**D**). At chronic phase, RA pressure trends to decrease. IVC: Inferior Vena Cava pressure curve. RA: Right Atrium pressure curve.

**Table 1 jcm-10-04601-t001:** Characteristics, advantages/disadvantages of percutaneous techniques for treating severe tricuspid regurgitation.

Technique	Favorable Conditions/Aspects	Unfavorable Conditions/Aspects
Edge-to-edge technique	Small Coaption gap (<7 mm) Antero- or postero-septal TR jet location RV lead without leaflet obstruction	Large coaptation gap (>10 mm) Antero-posterior TR jet location Poor echo windows RV lead with leaflet obstruction Diffusely degenerative/thickened leaflets Large learning curve Long procedural time
Annuloplasty	Favorable RCA location for safe annular anchoring Predominant RA dilatation driving annular dilatation	Large coaptation gap (>7 mm) Primary lead-induced TR Pulmonary hypertension Significant learning curve Long procedural time
Orthotopic Valve Replacement	Large coaptation gap (>7 mm) Primary leaflet pathology/organic TR Prior TV repair	RV dysfunction Severe pulmonary hypertension Very large annular dilatation (>52 mm) Contraindication for OAC
Heterotopic Valve Replacement	Anatomy that prohibits coaptation/annuloplasty/orthotopic TVR RV leads RV dysfunction Short learning curve Short procedural time General anesthesia/TEE not mandatory Back-up technique	IVC > 45 mm diameter Inadequate hepatic vein anatomy Very severe RV dysfunction Severe pulmonary hypertension Contraindication for OAC Higher risk of migration/embolization

OAC: Oral Anti-Coagulation. RCA: Right Coronary Artery. RV: Right Ventricle. TEE: Trans-Esophageal Echocardiography. TR: Tricuspid Regurgitation. TV: Tricuspid Valve.

**Table 2 jcm-10-04601-t002:** Compassionate use CAVI data.

Author	*N*	Device Used	Proc. Success	Embolisation	Leak	Open-Surgery	In-Hospital Death	30-Day Death
Lauten [23]	1	Customized	1 (100)	0 (0)	0 (0)	0 (0)	0 (0)	0 (0)
Laule [24]	3	Sapient XT	3 (100)	0 (0)	0 (0)	0 (0)	0 (0)	0 (0)
Lauten [25]	25	Sapien XT TricValve DirectFlow	23 (92)	2 (8)	NA	1 (4.0)	6 (24)	9 (36)
Toggweiler [7]	1	Tricento	1 (100)	0 (0)	0 (0)	0 (0)	0 (0)	0 (0)
Dreger [26]	14	Sapien XT	14 (100)	0 (0)	2 (14.3)	4 (28.6)	3 (21.4)	3 (21.4)
O’Neill [27]	24	Sapien XT	24 (100)	0 (0)	2 (10)	0 (0)	5 (20.8)	6 (25)
Cruz-Gonzalez [28]	6	Tricento	6 (100)	0 (0)	0 (0)	0 (0)	0 (0)	0 (0)
Aparisi [29]	2	TricValve	0 (0)	0 (0)	0 (0)	0 (0)	0 (0)	0 (0)
TricValve SP *	47	TricValve	46 (98)	1 (2)	NA	1 (2)	0 (0)	2 (8)

Data presented as *n* (%). * TricValve Special Program: data not published.

**Table 3 jcm-10-04601-t003:** Main clinical and hemodynamic eligibility criteria for the TRICUS Study and TRICUS EURO.

Inclusion Criteria
Clinical Severe symptomatic TR demonstrated by echocardiography with significant backflow in the IVC and/or SVC NYHA functional Class III or IV despite OMT LVEF ≥ 40% Distance covered in 6-min walk test ≥ 60 m The patient shall be screened by a “Heart Team” (including interventional cardiologist and cardiac surgeon) Hemodynamic Tricuspid v-wave ≥ 25 mmHg as demonstrated by right heart catheterization (measured in IVC and/or SVC Systolic arterial pulmonary pressure < 65 mmHg
Exclusion Criteria
Clinical Right ventricular failure (TAPSE ≤ 13 mmHg) Systolic pulmonary arterial pressure > 65 mmHg as assessed by Doppler echocardiography Untreated significant left sided valvular heart disease which requires treatment Requirement for other elective cardiac procedures (i.e., PCI, CABG, etc) Liver cirrhosis Child C Serum creatinine > 3.0 mg/dl or on dialysis Intra-cardiac shunt or congenital structural heart disease based on heart teams decision Documented primary coagulopathy or platelet disorder, or thrombocytopenia (absolute platelet count < 90 k) Contraindication or known allergy to device’s components, or VKA Thrombosis of the lower venous system or vena cava filter Life expectancy to less than one year for a non-cardiac condition

CABG: Coronary-Artery Bypass Graft. IVC: Inferior Vena Cava. LVEF: Left Ventricular Ejection Fraction. NYHA: New York Heart Association. OMT: Optimal Medical Therapy. PCI: Percutaneous Coronary Intervention. SVC: Superior Vena Cava. TR: Tricuspid Regurgitation. VKA: Vitamin K Antagonist.

**Table 4 jcm-10-04601-t004:** Anatomical criteria for currently available TricValve^®^ devices.

SVC Prosthesis Sizing	25 mm	29 mm	33 mm	
Confluence Innominate	Larger than 14 mm			
SVC to PA	19–31 mm	22–34 mm	25–40 mm	
SVC to middle PA	22–31 mm	27–34 mm	25–40 mm	
Length of middle PA	Larger than 35 mm			
Length of SVC to Confluence	Larger than 50 mm			
IVC prosthesis sizing	31 mm	35 mm	41 mm	45 mm
IVC-RA junction	24 to 31 mm	28 to 35 mm	33 to 41 mm	38 to 45 mm
IVC-on top of sHV confluence	24 to 31 mm	28 to 35 mm	33 to 41 mm	38 to 45 mm
Length IVC/RA Junction-sHV	Larger than 10 mm			
IVC just below sHV confluence	21 to 35 mm	27 to 43 mm	30 to 48 mm	35 to 50 mm
IVC at 5 cm below RA junction	21 to 35 mm	27 to 43 mm	30 to 48 mm	35 to 50 mm

IVC: Inferior vena cava. PA: Diameter of the superior vena cava at the level where right pulmonary artery crosses the svc. RA: Right Atrium. sHV: supra-Hepatic veins. SVC: Superior vena cava.

## Data Availability

Pubmed & European Commission of health technology and cosmetics.

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
