# Peer review of "Caval Valve Implantation (CAVI): An Emerging Therapy for Treating Severe Tricuspid Regurgitation"

_jcm, 2021, doi:10.3390/jcm10194601_

Round 1

Reviewer 1 Report

This is a very comprehensive review on the application of caval valves for severe tricuspid regurgitation. I found the topic very interesting, and the evidence was well presented. 

I have very few suggestions regarding the preparation of the manuscript. Firstly, I noted some possible spelling mistakes such as the "plain" in the figure legend of the Figure 1, rather than "plane".

Secondly, some expressions I found to be slightly ambiguous, such as the last sentence of the page 4: "However, despite this varied range of new devices, the minority of the overall severe TR population are ultimately deemed anatomically suitable for currently tested transcatheter therapies". Perhaps by changing the expression to "..., the majority of ...are ultimately deemed anatomically unsuitable..." would be easier to understand, in my humble opinion. In Table 1, in the row of the annuloplasty, I noted there is an expression of "unfavorable TR anatomy". I am not sure what it refers to, because I could see just above it, there is "large coaptation gap" which essentially is a feature of TR anatomy, however, there is lack of clarity of this "unfavorableness" in the context. I was wondering whether it could be briefly explained if applicable please. 

Thirdly, I was wondering whether a conclusion/summary could be added to the end of the manuscript to nicely summarize the evidence and suggest on future research directions if applicable please. 

Overall, it is a very informative paper and I learnt a lot reading through it as a cardiology clinician. Many thanks for your hardwork!

Author Response

Answer:

We thank the reviewers for their time, patience and positive feedback that has surely helped to improve the quality of the article. Most of suggestions of reviewers have been addressed in the reviewed version of the manuscript accordingly.

Specifically

Reviewed 1.

 I have very few suggestions regarding the preparation of the manuscript. Firstly, I noted some possible spelling mistakes such as the "plain" in the figure legend of the Figure 1, rather than "plane".

We have addressed this error in the reviewed version of the manuscript.

Secondly, some expressions I found to be slightly ambiguous, such as the last sentence of the page 4: "However, despite this varied range of new devices, the minority of the overall severe TR population are ultimately deemed anatomically suitable for currently tested transcatheter therapies". Perhaps by changing the expression to "..., the majority of ...are ultimately deemed anatomically unsuitable..." would be easier to understand, in my humble opinion.

We have changed the phrase accordingly with reviewer suggestion

In Table 1, in the row of the annuloplasty, I noted there is an expression of "unfavorable TR anatomy". I am not sure what it refers to, because I could see just above it, there is "large coaptation gap" which essentially is a feature of TR anatomy, however, there is lack of clarity of this "unfavorableness" in the context. I was wondering whether it could be briefly explained if applicable please. 

We agree with reviewer. This was a mistake that we have corrected it in the reviewed version of the manuscript.

Thirdly, I was wondering whether a conclusion/summary could be added to the end of the manuscript to nicely summarize the evidence and suggest on future research directions if applicable please. 

We have added a conclusion at the end of the reviewed version of the manuscript (see last paragraph of page 30).

“Initial data suggest that CAVI procedures with specific-designed devices are feasible and safe for treating severe TR. CAVI with TricValve® device is a relatively simply and predictable procedure, with less anatomical requirements than other tricuspid percutaneous devices, being a promising alternative for a wide range of patients affected of severe tricuspid regurgitation. However, its favorable effects from a safety and efficacy perspective still needs to be corroborated by the currently ongoing pivotal trials, and ultimately in a randomized clinical trial against optimal medical therapy”.

Reviewer 2 Report

Thank you for allowing me to review the manuscript. 

In this review, Dr. Abdul-Jawad Altisent nicely summarized all known on the CAVI technique for a specific subgroup of severe TR patients. Highlight the advantage and disadvantages of the method among all other treatments options. 

The review is well written and illustrated. I have no major comments.

I would suggest the following points to improve the quality of the paper. 

  1. Table 1- instead of coaptation - I would suggest "edge-to-edge technique"
  2. Figure 1 - I would recommend illustrating it better. I could not see all the details. It may be better to rephrase the related paragraph. too much confusion.   
  3. CAVI: anatomical consideration - "brachycephalic trunk" - I believe it should be a brachiocephalic trunk. Also, it is better to rephrase the paragraph to make it easy to understand. 
  4. Figures 2 and 3 - I would suggest improving the illustration and maybe adding reconstructed CT images. 
  5. I would suggest making to review shorter by creating a table summarizing the trials and devices.  
  6.  I suggest adding a short paragraph regarding IVC/SVC anomalies which may be important in the pre-procedural analysis and during the procedure. 

Good luck. 

Author Response

Answer

We thank the reviewers for their time, patience and positive feedback that has surely helped to improve the quality of the article. Most of suggestions of reviewer have been addressed in the reviewed version of the manuscript accordingly.

Table 1- instead of coaptation - I would suggest "edge-to-edge technique"

We have addressed this suggestion

Figure 1 - I would recommend illustrating it better. I could not see all the details. It may be better to rephrase the related paragraph. too much confusion.   

We have re-made and re-phrase figure 1 accordingly to reviewer suggestion by adding a CT scan reconstruction of central venous system. We hope that the revised version of figure 1 will be more understandable (Page 9, reviewed version of manuscript).

“A. Computed tomography (CT) scan reconstruction of central venous system, right heart and pulmonary artery at lateral plane. B. CT para-sagittal plane at the level of the vena cava. Superior vena cava (SVC) is usually 7.1 cm long, and has slight angulation (black line), usually at the level of right pulmonary artery (RPA) (Red line). Of note, the junction of SVC-right atrium (RA) and junction IVC-RA are not in the same anterior-posterior plane. Special caution applies to patients with short distance between the IVC-RA junction (green line) and inter-atrial-septum (*) owing to the risk of septal perforation during inferior caval prosthesis delivery. Eustachian valve (e) varies in size, may be very prominent. C. CT transversal plain at the level of RPA. Notable structures (i.e. aorta, trachea, pulmonary artery, etc) are adjacent but not attached to SVC minimizing risk of complications during CAVI. D. CT at para-sagittal plane of the IVC-RA junction showing the distance between hepatic vein (HV) confluence and IVC-RA junction (purple line). Higher distance at this level helps to stabilize the devices owing that it increases the contact surface between device-vein wall. The IVC-RA junction (yellow line) usually presents with a greater angulation than SVC-RA junction.

CAVI: anatomical consideration - "brachycephalic trunk" - I believe it should be a brachiocephalic trunk. Also, it is better to rephrase the paragraph to make it easy to understand. 

We have addressed and re-phrase the paragraph to make it easy to understand (page 8 and 9 of the reviewed version)

The SVC is 7.1 cm ±1.4 long and is formed by the union of the right and left brachiocephalic trunks (also called innominate); it returns the blood from the upper half of the body to the heart. It courses along the right middle mediastimun, with the trachea and ascending aorta on its left, drains over the anterior roof of the right atrium (RA), and does not have a valve. SVC is often irregular in shape. The Azygos vein is a major tributary and drains posteriorly into distal SVC. There are no other major tributaries, however SVC has several patterns of venous collaterals. Neither the SVC nor the brachiocephalic trunk contains valves. They do give rise to the internal, external and subclavian jugular veins, and their valves are usually located at the entrance to the subclavian and jugular veins. These vein valves can hinder the progression of catheters or wires. Other notable anatomical structures of the right atrium, such as the sinus node, are located in a lower level, in an area where CAVI devices should not be implanted (Figure 1 and 2)(20,21)”

Figures 2 and 3 - I would suggest improving the illustration and maybe adding reconstructed CT images. 

We have added a reconstructed CT images in Figure 1.

I would suggest making to review shorter by creating a table summarizing the trials and devices

We thank to reviewer 2 for the suggestion.

Table 2 is a summary of the trials and devices of CAVI published to date.

There is still no data published of the Pivotal Trials. Data from Pivotal Trials is expected to be published at the end of this year. For this reason we only include compassionate use CAVI data in the table.   

I suggest adding a short paragraph regarding IVC/SVC anomalies which may be important in the pre-procedural analysis and during the procedure. 

We have added a short paragraph explaining IVC and SVC anomalies.

Page 8, reviewed version.

Persistent left SVC is the most common congenital thoracic venous anomaly. Other congenital or acquired anomalies such as partial anomalous pulmonary venous return, SVC aneurysm or stricture may cause difficulty in CAVI procedures”.

Page 11, reviewed version of manuscript.

Congenital IVC variants are present in 4% of patients. The most common anomalies are the absence of IVC, duplication of the IVC (usually in infrarenal segments), left-side IVC, anomalous continuation of the suprarenal IVC as the Azygos or Hemiazygos vein, retrocaval ureter, IVC web formation (complete or fenestrated membrane in the intrahepatic IVC segment) and extra-hepatic porto-caval shunt (Abernethy malformation)”.